

# Transcriptomic analysis reveals ethylene signal transduction genes involved in pistil development of pumpkin

Qingfei Li, Li Zhang, Feifei Pan, Weili Guo, Bihua Chen, Helian Yang, Guangyin Wang and Xinzheng Li

College of Horticulture and Landscape, Henan Institute of Science and Technology, Xin Xiang, China
Henan Province Engineering Research Center of Horticultural Plant Resource Utilization and Germplasm Enhancement, Xin Xiang, China

Corresponding authors
Qingfei Li, lqf1988@hist.edu.cn, lqf_20131025@126.com
Xinzheng Li, liuzhw@hist.edu.cn, chenchao_20061225@126.com

## ABSTRACT

Development of female flowers is an important process that directly affects the yield of *Cucubits*. Little information is available on the sex determination and development of female flowers in pumpkin, a typical monoecious plant. In the present study, we used aborted and normal pistils of pumpkin for RNA-Seq analysis and determined the differentially expressed genes (DEGs) to gain insights into the molecular mechanism underlying pistil development in pumpkin. A total of 3,817 DEGs were identified, among which 1,341 were upregulated and 2,476 were downregulated. The results of transcriptome analysis were confirmed by real-time quantitative RT-PCR. KEGG enrichment analysis showed that the DEGs were significantly enriched in plant hormone signal transduction and phenylpropanoid biosynthesis pathway. Eighty-four DEGs were enriched in the plant hormone signal transduction pathway, which accounted for 12.54% of the significant DEGs, and most of them were annotated as predicted ethylene responsive or insensitive transcription factor genes. Furthermore, the expression levels of four ethylene signal transduction genes in different flower structures (female calyx, pistil, male calyx, stamen, leaf, and ovary) were investigated. The ethyleneresponsive DNA binding factor, *ERDBF3*, and ethylene responsive transcription factor, *ERTF10*, showed the highest expression in pistils and the lowest expression in stamens, and their expression levels were 78- and 162-times more than that in stamens, respectively. These results suggest that plant hormone signal transduction genes, especially ethylene signal transduction genes, play an important role in the development of pistils in pumpkin. Our study provides a theoretical basis for further understanding of the mechanism of regulation of ethylene signal transduction genes in pistil development and sex determination in pumpkin.

## INTRODUCTION

Pumpkin (*Cucurbita moschata* Duch.) is a typical monoecious plant with distinct male and female flowers. The number and proportion of female flowers can directly influence yields and economic benefits of *Cucurbitaceae* crops. The pistil is the main characteristic

structure of female flowers. It is well established that ethylene promotes pistil and female flower development. Application of ethylene, or inhibition of ethylene action, increases or decreases the number of pistil-bearing buds (*Papadopoulou et al., 2005*). Ethylene promotes carpel development and arrests stamen development in female flower (*Chen et al., 2016*). Treatment with exogenous ethylene or ethylene releasing reagents can increase the numbers of female and bisexual flowers in monoecious and andromonoecious lines, respectively (*Iwahori, Lyons & Sims, 1969*; *Malepszy & Niemirowicz-Szczytt, 1991*; *Manzano et al., 2011*). Auxin response factors (ARFs), an important component in auxin signalling pathway, especially the *ARF13* and *ARF17* genes are essential for pistil development in Japanese apricot (*Song et al., 2015*). The development of female flowers is inseparable from sex differentiation in *Cucurbitaceae* crops.

A number of studies have found that the floral development and sex expression in *Cucurbitaceae* crops can be affected by multiple phytohormones, including ethylene, auxin, cytokinin, gibberellin, abscisic acid, brassinosteroid, jasmonic acid, and salicylic acid (*Rudich & Halevy, 1974*; *Trebitsh, Rudich & Riov, 1987*; *Yamasaki, Fujii & Takahashi, 2003*; *Menéndez et al., 2009*; *Pimenta Lange & Lange, 2016*; *Mao et al., 2017*; *Zhang et al., 2017*). Ethylene might be the major hormone in sex determination. In cucumber, sex differentiation is mainly determined by the *F* (*CsACS1G*), *M* (*CsACS2*), and *A* (*CsACS11*) genes. The *F*, *M*, and *A* genes encode 1-aminocyclopropane-1-carboxylate (ACC) synthase (ACS), which is a key rate-limiting enzyme in ethylene biosynthetic pathway (*Pierce & Wehner, 1990*; *Pan et al., 2018*). Among these, the *F* gene promotes the development of female flowers (*Mibus & Tatlioglu, 2004*; *Knopf & Trebitsh, 2006*), and the *M* gene inhibits the development of stamens (*Yamasaki et al., 2001*; *Saito et al., 2007*). The *CsACS11* (*A*) is an androecious gene, and mutants with loss of *CsACS11* function were found to be androecious, with no female flowers (*Boualem et al., 2015*). The ACC oxidase gene (*ACO*), another key gene in ethylene biosynthesis, is also essential for the development of female flowers. A transcription factor gene, *CsWIP1*, could directly bind the promoter of *CsACO_2* to repress its expression (*Chen et al., 2016*). In melon, sex determination is governed by the genes, andromonoecious (a) and gynoecious (g). *CmACS-7* is the andromonoecious gene (*Boualem et al., 2008*). *CmACS11* controls the development of female flowers in melon and its function is not exactly the same as of *CsACS11*. *CmWIP1*, the ortholog of *CsWIP1*, can negatively regulate the femaleness and gynoecious plants are obtained when there is a loss of function of *CmWIP1*. The expression patterns of *CmWIP1* and *CmACS11* are opposite in melon. *CmACS11* can repress the expression of *CmWIP1* at an upstream step in the sex determination pathway to control flower development. Ethylene could inhibit stamen development through *CmACS-7* and *CmACS11* (*Boualem et al., 2015*). Thus, the sex differentiation and floral development in melon could be a result of the interaction among *CmACS11*, *CmWIP1*, and *CmACS-7*.

Besides the genes involved in ethylene biosynthesis, those involved in ethylene signal transduction have also been implicated in the development of female flowers in cucumber. Previous studies have revealed that there existed organ-specific DNA damage in primordial anther of female flowers and that the DNA damage was induced via the ethylene signaling pathway. The ethylene-receptor gene, *CsETR1*, located in the pistil primordia has also

been involved in the arrest of stamen development through induction of DNA damage in female flowers (*Hao et al., 2003*; *Yamasaki, Fujii & Takahashi, 2003*; *Duan et al., 2008*; *Wang et al., 2010*). It can be bound and activated by *CsAP3* in vitro and in vivo (*Sun et al., 2016*). Moreover, the expressions of $C_S$-*ETR2* and $C_S$-*ERS* were reported to be regulated by ethylene, because their mRNAs were significantly elevated by the application of ethrel and their levels were lowered by the application of an ethylene inhibitor, aminoethoxyvinyl glycine (*Yamasaki, Fujii & Takahashi, 2000*). Thus, not only the genes involved in ethylene synthesis, those involved in ethylene-mediated signal transduction contribute to the development of female flowers. However, the evidence for how the ethylene signal transduction genes regulate the development of female flowers is still weak.

Besides ethylene, other phytohormones are also involved in the development of female and male flowers. Gibberellins (GA) can promote the male tendency. Its production in andromonoecious cucumber is higher than that in monoecious and gynoecious cucumber (*Hemphill, Baker & Sell, 1972*). A recent report indicated that GA could regulate sex expression via ethylene-dependent and ethylene-independent pathways (*Zhang et al., 2017*). The application of indole-3-acetic acid (IAA), except for the Beta -Alfa type, could enhance ethylene and ACC production (*Trebitsh, Rudich & Riov, 1987*). Cytokinins are possibly involved in determining the morphological differences between sex types. The endogenous levels of the cytokinins were found to be higher in female gametophytes than in male gametophytes (*Menéndez et al., 2009*). Abscisic acid could promote the female tendency of gynoecious plants. It participates in the sex regulation by inhibiting the GA activity of cucumber (*Rudich & Halevy, 1974*). Jasmonic acid signaling also plays an important role in flower development in plants, especially in stamen sterility, sex determination, female flower development, and seed maturation (*Yuan & Zhang, 2015*; *Mao et al., 2017*).

As evident from the above studies, floral development and sex differentiation is a result of the interaction of various plant hormones and ethylene apparently plays a key role in these processes. However, little is known about how the plant hormone signal transduction genes are involved in female and male flower development. Herein, to explore the key genes involved in the development of pistils, the aborted and normal pistils of pumpkin were used for RNA-Seq analysis. The results showed that plant hormone signal transduction genes, especially ethylene related genes, play important roles in the development of pistils in pumpkin. Fourteen DEGs, which were mostly annotated as plant hormone signal transduction genes, were chosen for qRT-PCR verification. Furthermore, four ethylene signal transduction genes were appealing candidates for investigation of their expression in different flower structures. Our results provide a foundation for dissecting the molecular mechanism of regulation of pistil formation in pumpkin by the candidate genes, identified herein.

# MATERIAL AND METHODS

## Plant materials

The aborted plants were from the same inbred line as the normal plants and exhibited similar plant characters except for the female development. The seeds of pumpkin were

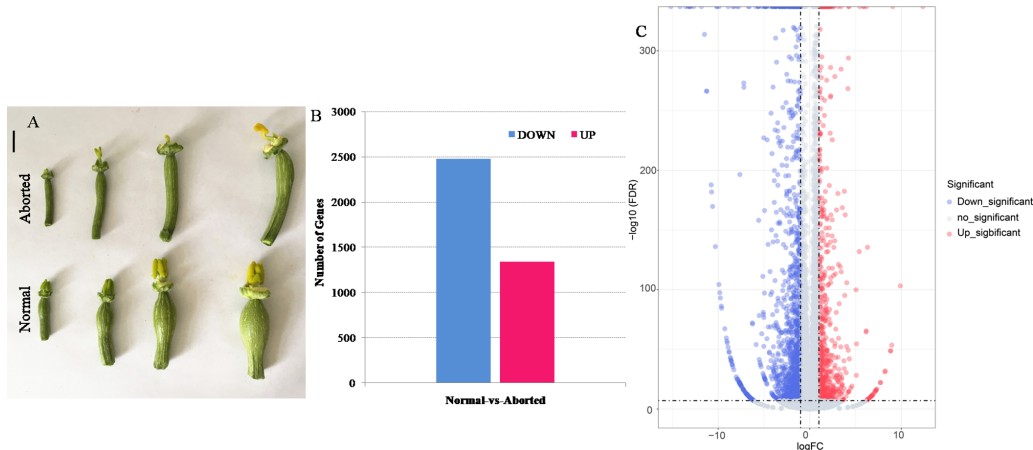

**Figure 1** **The aborted and normal pistils and differentially expressed genes (DEGs) that were significantly upregulated and downregulated between the aborted and normal pistils of pumpkin.** (A) The aborted and normal pistils (bars = 1 cm). (B) The number of upregulated and downregulated DEGs. (C) Volcano map for the gene expression. The |log2 (fold change)| > 1 and FDR <0.05 were used as the cut-offs for significance.

sterilized with hot water (55 °C) for approximately 15 min, and then soaked in water for 4 h. The seeds were germinated on a wet filter paper in a Petri dish at 28 °C in the dark. When at least 80% of the seeds had germinated, they were transferred into a substrate mixture of peat, vermiculite, and perlite (3:1:0.5, v/v) and grown in an artificial climate room under conditions of 14-h light/10-h dark and day/night temperatures of 25 °C/18 °C. When the seedlings had developed three true leaves, they were transplanted to a field and grown under the natural environment. At flower development period, the aborted female flowers exhibited different from those of the normal plants. When the length of corolla of flower bud was less than five mm, there was no difference between the aborted female flower and the normal one. As the buds grow bigger, in aborted pistils the incomplete stigma can be seen when the corolla is peeled off, and ovary looks thinner than in normal pistils, which is as shown in Fig. 1A. Thus, the aborted and normal pistils, which were come from the female flower buds with five mm length of corolla and located in the shoot apices, were used for RNA-Seq analysis and for the validation of gene expression. Moreover, leaf, female calyx, pistil, ovary, male calyx, and stamen from the male and female flowers of the same plants, were used to detect the expression of the candidate genes in different floral structures. All the samples were immediately frozen in liquid nitrogen and stored at −80 ° C for further experiments.

## RNA isolation

Total RNA was extracted using the Trizol$^{TM}$ reagent (Invitrogen, Carlsbad, CA, USA), as described by the manufacturer. DNase I was then used to remove DNA contamination in the RNA preparations. The purity and integrity of RNA samples were evaluated by electrophoresis on 1% RNase-free agarose gels, spectrophotometry (A260/A280 and

A260/A230) on NanoDrop DU8000, and analysis using the Agilent 2100 Bioanalyzer system (Agilent, USA).

## Construction and sequencing of cDNA libraries

The NEBNext Ultra Directional RNA Library Prep Kit for Illumina (NEB, Ispawich, USA) was used for mRNA fragmentation. In brief, poly (A) mRNA was extracted from total RNA using oligo-dT magnetic beads, and further fragmented. First strand cDNA was synthesized using random hexamer primers and reverse transcriptase. Second strand cDNA was synthesized using dNTPs, DNA polymerase I, RNase H, and the first strand. To select the fragments that were preferentially 150–200 bp in length, the cDNA fragments were purified with an AMPure XP system (Beckman Coulter, USA). The high-quality libraries were then sequenced on the Illumina HiSeq 4000 platform by Sagene Co. (Guangzhou, China). The libraries of two biological replicates of aborted and normal pistils were prepared independently. The raw transcriptome data were deposited in the National Center for Biotechnology Information Sequence Read Archive under BioProject number PRJNA554766.

## Bioinformatics analysis of differentially expressed genes (DEGs) data

The raw reads were pre-processed by removing the adaptor sequences, low-quality reads (more than 50% bases with SQ $\leq$20 in one sequence), and reads with more than 5% N bases (bases unknown). The clean reads were mapped to the *Cucurbita moschata* genome (http://cucurbitgenomics.org/organism/9) using TopHat2 v2.1.1 (*Trapnell et al., 2012*; *Kim et al., 2013*; *Sun et al., 2017*), allowing up to two mismatch. Functional annotation was performed by searching against the non redundant (NR), Swiss–Prot and clusters of orthologous groups for eukaryotic complete genomes (KOG) databases using BLAST with an *E*-value of 1e$^{-5}$. The R package edgeR was used to identify the DEGs using raw read counts as input data (*Robinson, McCarthy & Smyth, 2010*). *P*-values were adjusted using Benjamini and Hochberg's method to control the false discovery rate (FDR) (*Benjamini & Hochberg, 1995*). The |log2 (fold change)|> 1 and adjusted *P*-value < 0.05 for multiple tests using the Benjamini method were used as significance cut-offs for the expression differences.

## Functional enrichment analysis for DEGs

Gene functional enrichment analysis of DEGs was implemented using the GOSeq R package (*Young et al., 2010*). GO terms included cellular component, molecular function, and biological process. After the hypergeometric test, Bonferroni correction was employed for *P*-value correction with a cut-off of 0.05. The GO terms satisfying the condition were considered significantly enriched by DEGs. Furthermore, the Kyoto Encyclopedia of Genes and Genomes (KEGG) enrichment analysis was performed. KEGG is the main public database of pathways. FDR control method was used to identify the threshold of the *P*-value in multiple tests (*Benjamini & Hochberg, 1995*). Pathways with their Benjamini and Hochberg adjusted *P*-values $\leq$0.05 were defined as significantly enriched by DEGs (*Mao et al., 2005*; *Kanehisa et al., 2014*). Using significant enrichment of pathways, the
**Table 1** Statistical results of reads mapped to reference genome and expressed genes in different libraries.

| Sample | Total pair reads | Mapped pair reads (Ratio) | Known gene num | Novel gene num |
|--------|------------------|---------------------------|----------------|----------------|
| A2 | 55729080 | 49972699(89.67%) | 26128 | 386 |
| A4 | 65402194 | 58874465(90.02%) | 26463 | 386 |
| N2 | 58025908 | 51001759(87.89%) | 26559 | 381 |
| N4 | 56371344 | 50670138(89.89%) | 25914 | 372 |

major biochemical metabolic pathways and signal transduction pathways involving DEGs can be determined.

## Quantitative real-time RT-PCR

RNA extraction and detection were done as described above. cDNA synthesis was conducted using the PrimeScript$^{TM}$ RT Master Mix (Perfect Real Time) Reagent Kit (Takara, Dalian, China), according to the manufacturer's instructions. The gene primers used in the quantitative real-time RT-PCR (qRT-PCR) experiment are listed in Table S1. *ACTIN* was used as an internal control. The qRT-PCR system was consisted of 10 μL SYBR Premix Ex Taq$^{TM}$ II, 0.8 μL each primer (10 μM), 2 μL diluted cDNA (150 ng), and 6.4 μL nuclease-free water. The qRT-PCR reactions were performed in three biological replicates. The qRT-PCR was carried out using a Bio-Rad IQ5 instrument (Foster City, CA, USA), with the following conditions: 1 cycle of 95 °C for 40 s, 40 cycles of 95 °C for 5 s and 61 °C for 30 s. The relative expression levels were calculated using the $2^{-\Delta\Delta Ct}$ method (*Livak & SchmittgenT, 2001*). To verify the RNA-Seq data, the correlation analysis and the Pearson correlation coefficient between log2 (fold change) of RNA-Seq and qRT-PCR was calculated using the IBM SPSS statistics 22 software.

## RESULTS

### Summary of the sequencing data

After filtering, 5.62, 6.60, 5.86, and 5.72 million high quality clean reads for aborted (A2, A4) and normal (N2, N4) pistils were obtained. Among these high quality clean reads, the percentages of Q20 were 97.46%, 97.58%, 96.61%, and 97.45%, respectively, and the percentage of GC were 45.55%, 45.49%, 45.77%, and 45.81%, respectively. The total pair reads, mapped pair reads (ratio) of these libraries according to the *Cucurbita moschata* genome, known gene number, and number of novel genes were as shown in Table 1. Finally, 3,817 DEGs were identified, including 1,341 up-regulated and 2,476 down-regulated genes, in the aborted pistils compared with the normal pistils (Figs. 1B, 1C, Table S2).

### GO enrichment analysis

The DEGs were enriched in cellular component, molecular function, and biological process groups. A total of 749 DEGs were categorized into biological process category, which mainly included metabolic process (213 up-regulated, 345 down-regulated), cellular process (179 up-regulated, 307 down-regulated), and single-organism process (151 up-regulated, 235 down-regulated) (Fig. 2). Among these, the most significantly enriched GO terms

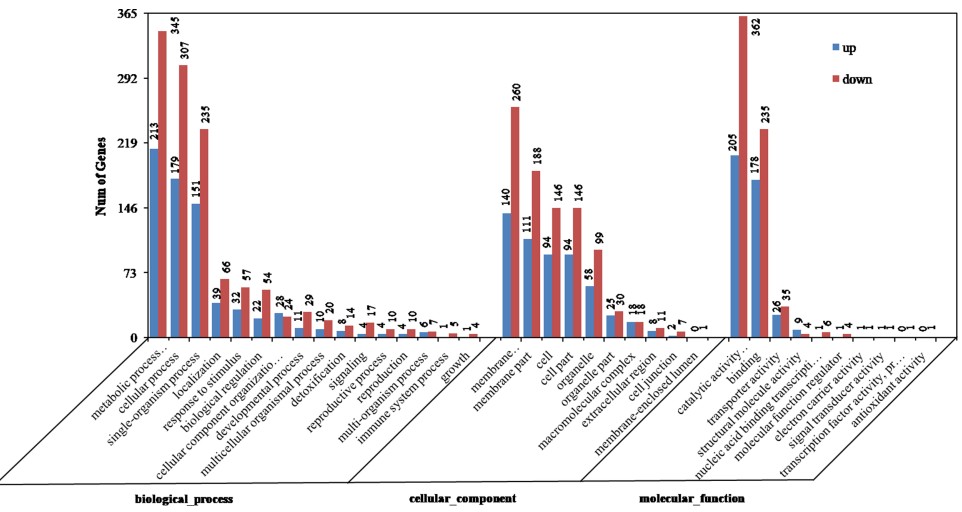

**Figure 2  Gene ontology (GO) categories of differentially expressed genes (DEGs) between the aborted and normal pistils.**

included single-organism metabolic process (256, $P = 9.21E−12$), hydrogen peroxide metabolic process (16, $P = 7.80E−07$), phenylpropanoid metabolic process (10, $P = 4.34E−04$), and carbohydrate metabolic process (49, $P = 4.41E−04$). 580 DEGs were divided into the cellular component category, and they mainly enriched in membrane (140 up-regulated, 260 down-regulated), membrane part (111 up-regulated, 188 down-regulated), and cell (94 up-regulated, 146 down-regulated) terms (Fig. 2). The most significantly enriched GO terms in the cellular component category included extracellular region (19, $P = 5.01E−05$), membrane (400, $P = 3.94E−04$), photosystem II (6, $P = 5.37E−03$), and external encapsulating structure (23, $P = 9.34E−03$). A total of 762 DEGs were divided into molecular function, which mainly enriched in the catalytic activity (205 up-regulated, 362 down-regulated), binding (178 up-regulated, 235 down-regulated), and transporter activity (26 up-regulated, 35 down-regulated). The most significantly enriched GO terms in the molecular function category included oxidoreductase activity (152, $P = 1.47E−17$), tetrapyrrole binding (56, $P = 1.69E−12$), ion binding (186, $P = 6.80E−05$), and cation binding (167, $P = 1.96E−04$) (Fig. 2). On the whole, in cellular component, molecular function, and biological process groups, the down-regulated genes were all more than up-regulated ones.

## Plant hormone signal transduction and phenylpropanoid biosynthesis involved in pistil development in pumpkin

The KEGG pathway enrichment analysis indicated that the DEGs were significantly enriched in the following pathways: phenylpropanoid biosynthesis, plant hormone signal transduction, phenylalanine metabolism, tryptophan metabolism, linoleic acid metabolism (Fig. 3). It showed that 84 DEGs were enriched in the plant hormone signal transduction pathway (Table S3), which accounted for 12.54% of all the pathway annotated DEGs. The plant hormones involved in this pathway included ethylene, auxin, cytokinin, gibberellin,

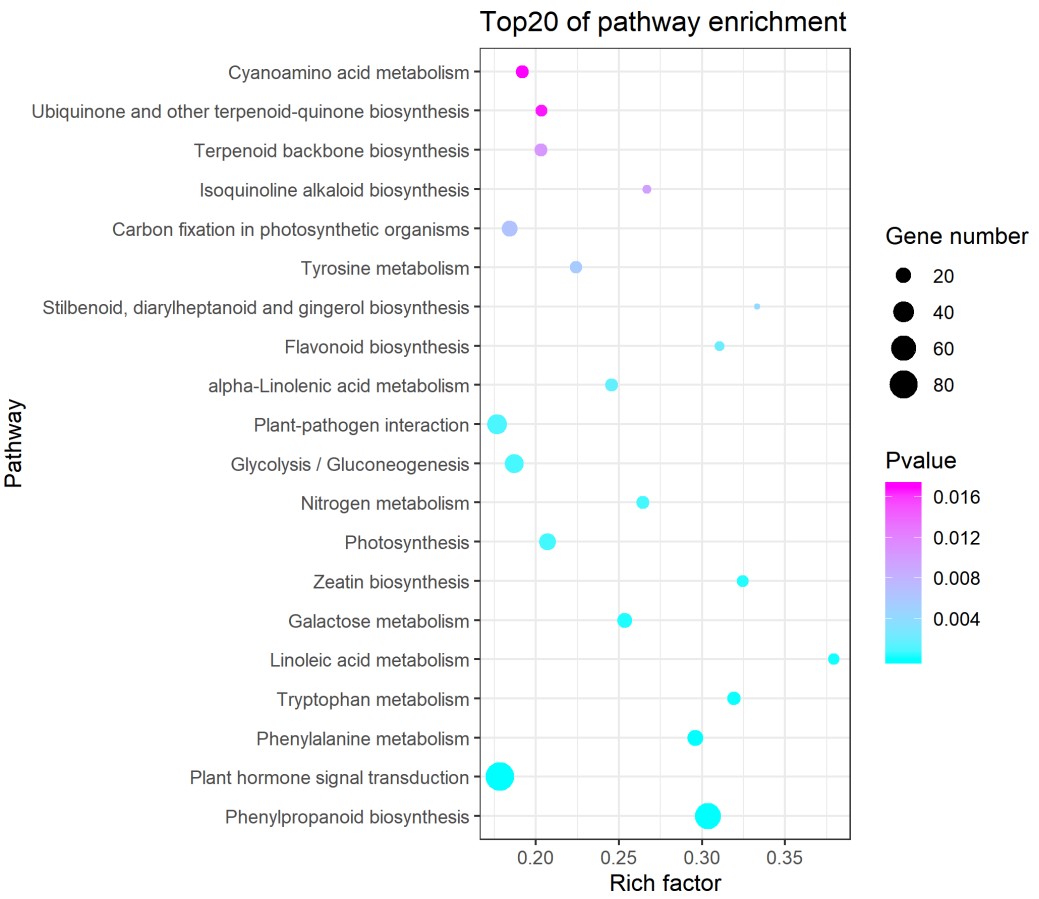

**Figure 3 Kyoto Encyclopedia of Genes and Genomes (KEGG) pathway enrichment of differentially expressed genes (DEGs) between the aborted and normal pistils.** Ggplot2 package for R language was used to generate this figure. The bigger the bubble, the more the DEGs. The smaller the pvalue, the more significant the KEGG enrichment. Rich Factor represents the ratio of DEGs number in the pathway to the total number of genes in the pathway. The larger the RichFactor, the higher the enrichment.

**Table 2 List of *PAL* in the significant DEGs.**

| Gene ID | Gene annotation | Fold change | FDR |
|---|---|---|---|
| CmoCh03G012270 | PREDICTED: phenylalanine ammonia-lyase-like [*Cucumis melo* ] | 0.439562 | 3.20E−05 |
| CmoCh03G012290 | PREDICTED: phenylalanine ammonia-lyase-like [*Cucumis melo* ] | 0.343162 | 1.13E−28 |
| CmoCh07G009470 | phenylalanine ammonia-lyase 3 [*Luffa aegyptiaca* ] | 0.458279 | 4.76E−08 |
| CmoCh07G009540 | phenylalanine ammonia-lyase 3 [*Luffa aegyptiaca* ] | 0.077247 | 1.86E−26 |
| CmoCh07G009550 | phenylalanine ammonia-lyase 3 [*Luffa aegyptiaca* ] | 0.209537 | 1.47E−13 |
| CmoCh07G009560 | PREDICTED: phenylalanine ammonia-lyase-like [*Cucumis melo* ] | 0.368197 | 4.93E−05 |
| CmoCh20G005320 | phenylalanine ammonia-lyase 4 [*Luffa aegyptiaca* ] | 2.000495 | 0 |

abscisic acid, brassinosteroid, jasmonic acid, and salicylic acid. Most of DEGs enriched in the plant hormone signal transduction pathway were annotated as predicated ethylene-responsive transcription factor, ethylene insensitive transcription factor, and ethylene receptor genes in NCBI blast and *Cucurbita moschata* genome (Table S4, Fig. S1). In addition, we found that 68 DEGs were enriched in phenylpropanoid biosynthesis (Table S5), which accounted for 10.15% of all the pathway annotated DEGs. Among these, seven DEGs were defined as phenylalanine ammonia-lyase genes (*PAL*), which coded for the key first rate-limiting enzyme in the phenylpropanoid pathway. It is worth noting that six *PAL* DEGs were downregulated in the aborted pistils (Table 2). The results indicated that the phenylpropanoid pathway, especially *PAL*, might be implicated in pistil development in pumpkin.

## Verification of RNA-Seq data by qRT-PCR analyses

The RNA-Seq results were validated by qRT-PCR assays. Fourteen annotated DEGs were chosen for qRT-PCR analyses. Twelve of these genes were downregulated and two were upregulated. It included auxin related genes, *AUX-IAA* (CmoCh04G015610, auxin-responsive protein IAA gene) and *ARF* (CmoCh16G005850, auxin response factor), gibberellins-related gene, *TF* (CmoCh05G010630, PIF4: phytochrome-interacting factor 4), ethylene related genes, *ACO* (CmoCh02G000640, 1-aminocyclopropane-1-carboxylate oxidation), *ETR* (CmoCh08G004320, ethylene receptor), *ERDBF3* (CmoCh17G005350, ethylene response DNA binding factor), *ERTF10* (CmoCh11G005450, ethylene responsive transcription factor gene), and *AP2* (CmoCh01G000430, ethylene responsive transcription factor gene), brassinosteroid biosynthesis genes, *TCH4* (CmoCh11G004020, xyloglucan:xyloglucosyl transferase) and *CYCD3* (CmoCh01G018690, cyclin D3), jasmonic acid pathway gene, *JAR1* (CmoCh06G010560, jasmonic acid-amino synthetase), salicylic acid pathway gene, *NPR1* (CmoCh03G001860, regulatory protein), and transcription factor, *TGA* (CmoCh13G003300), and phenylpropanoid biosynthesis gene, *PAL* (CmoCh07G009540, phenylalanine ammonia-lyase). As shown in Fig. 4, all the detected genes showed similar expression trends in qRT-PCR analyses as in RNA-Seq data, with a relative coefficient of $R^2 = 0.7887$ (Figs. 4A, 4B). The Pearson correlation analysis indicated that the RNA-Seq and qRT-PCR were strongly correlated ($R = 0.888$, $P = 0.00002$).

## Expression levels of ethylene candidate genes in different flower structures

Ethylene signal transduction genes (*ERDBF3*, *ERTF10*, *AP2*, and *ETR*) were appealing candidates for investigating their gene expression in different flower structures, including pistil, female calyx, ovary, stamen, male calyx, and leaf. The results showed that the expression level of ethylene response DNA binding factor, *ERDBF3*, was highest in pistil, followed by that in male calyx, leaf, female calyx, ovary, and stamen (Fig. 5), and the expression level in pistil was 78-times more than that in stamen. The expression level of *ERTF10* was also highest in pistil, followed by that in leaf, male calyx, female calyx, ovary, and stamen (Fig. 5), and the expression level in pistil was 162-times more than that in stamen. *ERTF10* and *ERDBF3* showed the highest expression in pistil and the lowest

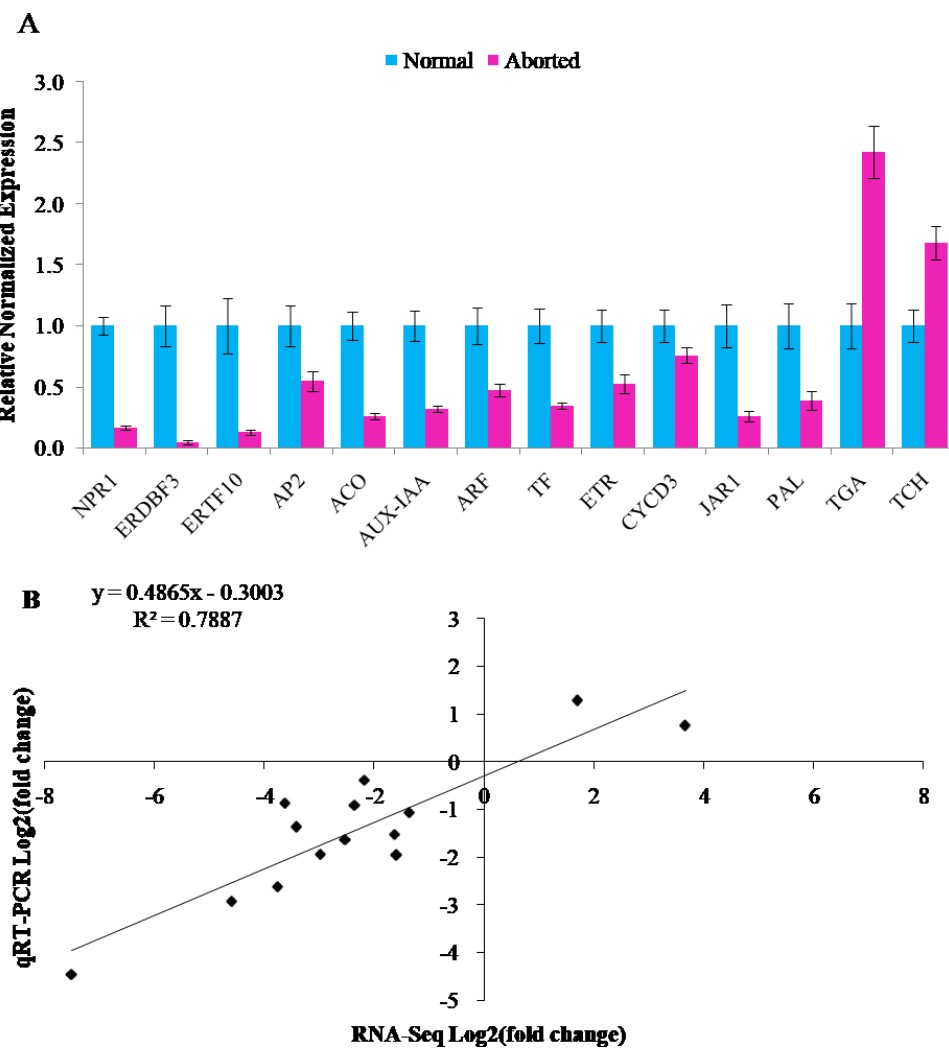

**Figure 4** **Verification of RNA-Seq Data by quantitative real-time PCR.** (A) Relative normalized expression of 14 selected genes in the normal and aborted pistils using qRT-PCR. Error bars indicate the standard errors. *AUX-IAA* (auxin-responsive protein IAA), *ARF* (auxin response factor), *TF* (*PIF4*, phytochrome-interacting factor 4), *ACO* (1-aminocyclopropane-1-carboxylate oxidation), *ETR* (ethylene receptor gene), *ERDBF3* (ethyleneresponse DNA binding factor 3), *ERTF10* (ethylene responsive transcription factor 10), *AP2* (like ethylene-responsive transcription factor), *TCH4* (xyloglucan endotransglucosylase/hydrolase), *CYCD3* (cyclin D3), *JAR1* (jasmonic acid-amino synthetase), *NPR1* (BTB/POZ domain and ankyrin repeat-containing protein NOOT2), *TGA* (TGACG-sequence-specific DNA-binding protein), *PAL* (phenylalanine ammonia-lyase). (B) Comparison of the expression ratios of 14 selected genes in using RNA-seq and qRT-PCR.

in stamen. *AP2* has been reported in other crops to be important for flower and seed development (*Jofuku et al., 1994*; *Zhang et al., 2018*). In this study, the expression level of *AP2* was found to be higher in the reproductive organ stamen, male calyx, female calyx, pistil, and ovary compared to that in leaf (Fig. 5). This indicates that, in pumpkin, *AP2* may also play an important role in flower development. The expression level of the ethylene receptor, *ETR*, was highest in stamen, followed by that in leaf, ovary, female calyx, pistil, and

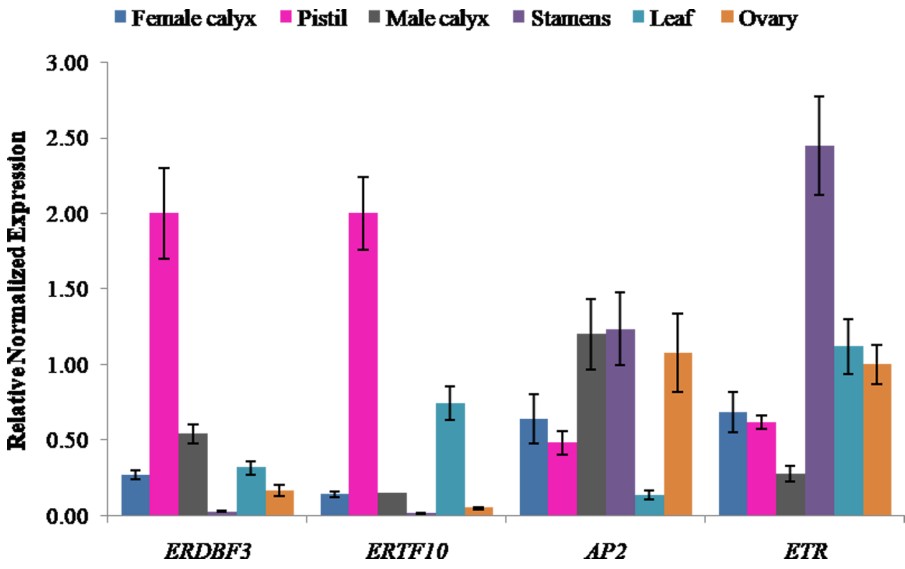

**Figure 5 Expression levels of ethylene signal transduction candidate genes (*ERDBF3*, *ERTF10*, *AP2*, and *ETR*) in different flower structures.** Error bars indicate the standard errors. *ERDBF3* (ethyleneresponse DNA binding factor 3), *ERTF10* (ethylene responsive transcription factor 10), *AP2* (like ethylene-responsive transcription factor), *ETR* (ethylene receptor gene).

male calyx (Fig. 5). From the results, we can see that the four ethylene signal transduction genes are important for the development of female and male flowers, and *ERTF10* and *ERDBF3* may especially be more important in the pistils of female flowers than that in the stamens of male flowers.

## DISCUSSION

The number and proportion of female flowers are directly related to the yield of *Cucurbitaceae* crops. In previous studies, researchers have focused on sex differentiation to explore the key genes and mechanisms regulating the development of female and male flowers. Ethylene is known to promote the development of female flowers, which was mainly manifested in sexual expression, the earliness and the larger number of female flowers per plant. Moreover, the development of female flowers requires much more ethylene than the development of male flowers (*Pierce & Wehner, 1990*; *Malepszy & Niemirowicz-Szczytt, 1991*; *Manzano et al., 2011*; *Manzano et al., 2014*; *Pan et al., 2018*).

In this study, the key genes involved in pistil development were studied directly using aborted pistil materials. The results of KEGG pathway analysis showed that the DEGs were enriched in plant hormone signal transduction pathway, and most of them were annotated as predicated ethylene-responsive transcription factor, ethylene insensitive transcription factor, and ethylene receptor genes (Table S5). This result echo a previous study which have suggested that ethylene synthesis and signal transduction play important roles in sex expression of pumpkin (*Cucurbita maxima*) (*Wang et al., 2019*). Furthermore, we found that the ethylene response DNA binding factor, *ERDBF3*, and the ethylene responsive

transcription factor, *ERTF10*, showed the highest expression in pistils and the lowest expression in stamens (Fig. 5). These results indicate that ethylene signal transduction genes, *ERTF10* and *ERDBF3*, may be more important in the pistils of female flowers than in other floral structures. In addition, previous studies have also shown that the ethylene receptor, *CsETR1*, and the ethylene-responsive transcription factor, *CsAP3*, are involved in the development of female flowers (*Hao et al., 2003*; *Yamasaki, Fujii & Takahashi, 2003*; *Wang et al., 2010*; *Sun et al., 2016*). The results presented in Fig. 5 show that *AP2* and *ETR* were expressed in all the structures of flowers that were assessed, and although their expression level in the stamens was higher than that in the pistils, there is not enough evidence to show as to how important they are for the development of female and male flowers. Except for the ethylene signal transduction, which might play a major role in the development of female flowers, the results of KEGG analysis also indicated that other plant hormone signal transduction pathways, including auxin, cytokinin, gibberellin, abscisic acid, brassinosteroid, jasmonic acid, salicylic acid signaling pathways, are involved in flower development. There is a possibility for the existence of interactions between various plant hormones through their signal transduction pathways. For example, ethylene synthesis can be induced by auxin, gibberellin, and jasmonic acid (*Rudich & Halevy, 1974*; *Trebitsh, Rudich & Riov, 1987*; *Zhang et al., 2017*; *Schubert et al., 2019*).

Phenylpropanoids comprise many aromatic metabolites, including the cell wall structural component, lignin, and many small phenolic molecules, such as coumarins, stilbenes, flavonoids, anthocyanins, and condensed tannins (*Vogt, 2010*; *Fraser & Chapple, 2011*; *Zhang, Gou & Liu, 2013*). Phenylalanine ammonia-lyase catalyzes the first rate-limiting step in the phenylpropanoid pathway, which controls the carbon flux to aromatic compounds, and to lignin. The results of this study indicate that six *PAL* DEGs are downregulated in aborted pistils (Table 2). Previous study have indicated that transgenic tobacco plants carrying antisense and sense *pal* cDNAs resulted in partial male sterility, with the reduction of pollen fertility (*Matsuda et al., 1996*). At present, there has been no systematic research on the mechanisms through which phenylpropanoids take part in the development of female flowers.

In addition, the DEGs in the present study were obtained from RNA-Seq analysis of the aborted and normal pistils, which were come from the flower buds with five mm length of corolla. More genes related to pistil development were mined. But, when the aborted female flower buds grew bigger, they exhibited incomplete stigma and thinner ovary for the absence of ovules, and which is the primary cause of pistil abortion is unclear. Further gene function analysis is needed to illuminate the mechanism of pistil abortion.

## CONCLUSIONS

In the present study, DEGs in aborted and normal pistils of pumpkin were identified. KEGG enrichment analysis showed that the DEGs were significantly enriched in plant hormone signal transduction, which accounted for 12.54% of the significant DEGs. Most of them were annotated as predicated ethylene signal transduction genes. The expression analysis of *ERDBF3* and *ERTF10* in different flower structures showed the highest expression in

pistils and the lowest expression in stamens. These results suggest that plant hormone signal transduction genes, especially ethylene signal transduction genes, play an important role in the development of pistils in pumpkin.

### Funding

This research was supported by the Key Science and Technology Program of Henan Province (No. 182102110049 and 192102110155), the Henan Province Construction Project for Bulk Vegetable Industry Technical System (No. S2010-03-G06), and the PhD research startup foundation of the Henan Institute of Science and Technology. The funders had no role in study design, data collection and analysis, decision to publish, or preparation of the manuscript.

### Grant Disclosures

The following grant information was disclosed by the authors:
Key Science and Technology Program of Henan Province: 182102110049, 192102110155.
Henan Province Construction Project for Bulk Vegetable Industry Technical System: S2010-03-G06.
PhD Research Startup Foundation of Henan Institute of Science and Technology.

### Competing Interests

The authors declare there are no competing interests.

### Author Contributions

- Qingfei Li conceived and designed the experiments, performed the experiments, prepared figures and/or tables, and approved the final draft.
- Li Zhang performed the experiments, prepared figures and/or tables, and approved the final draft.
- Feifei Pan analyzed the data, authored or reviewed drafts of the paper, and approved the final draft.
- Weili Guo, Bihua Chen and Helian Yang analyzed the data, prepared figures and/or tables, and approved the final draft.
- Guangyin Wang and Xinzheng Li conceived and designed the experiments, authored or reviewed drafts of the paper, and approved the final draft.

### Data Availability

   Data is available at NCBI BioProject: PRJNA554766.
   The data is also available at Figshare: Li (2020): Transcription analysis of aborted pistils in pumpkin. figshare. Dataset. https://doi.org/10.6084/m9.figshare.11515167.v1.

### Supplemental Information

Supplemental information for this article can be found online at http://dx.doi.org/10.7717/peerj.9677#supplemental-information.

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
