# Peer review of "Transcriptomic analysis reveals ethylene signal transduction genes involved in pistil development of pumpkin"

_PeerJ, doi:10.7717/peerj.9677_

## Round 0.1 · original submission · Major Revisions

1) All three reviewers comment about your raw data and supplemental tables. Please ensure that the data you discuss are complete and make them available to the readers.
2) The figures must be improved to be readable (adequate font size and friendly to colour-blinded).
3) Several conclusions about DEGs would need more documentation and/or experimental support and discussion (e.g. effect of ethylene on development).

·

Basic reporting

This manuscript by Li et al. reports on an RNA-seq study related to pistil development in pumpkin. The authors find that especially genes related to ethylene signaling are enriched, when comparing pistils and aborted pistils. A few genes are verified by qRT-PCR. A drawback is that the RNA-seq is only performed in duplicate and not triplicate, which is the current standard.
It has been known that ethylene affects flower and fruit development. For instance, many studies reported on the effect of ethylene in cucumber, which the authors also cite.
The manuscript reads well and the figures are clear. The data will be interesting for people working with pumpkin.

Experimental design

In general good, however, a drawback is that the RNA-seq is only performed in duplicate and not triplicate, which is the current standard.

Validity of the findings

No comment.

Additional comments

Comments to the authors:
1. In pumpkin, a recent study reported also an RNA-seq study (Wang et al. 2019, doi:10.3390/ijms20133185) and conclude that ethylene and auxin signaling is enriched, and that this affects sex termination of the flowers. This study has not been cited, though the study is not the same but related, and the authors should cite this study and discuss their results in the light of the published data.
2. It would have been nice to add some experiments applying for instance ethylene to developing plants and observe the effect on flower and pistil development.
3. The authors could add some more visualization of their data (eg. heatmap) related to the ethylene pathway.
4. Fig. 4 The authors write that Pearson correlation was calculated, but it would be good to see per gene the correlation between qRT-PCR and RNA-seq in the same figure.
5. Suppl. Table 2. Add accession numbers of the homologues genes in the column ´Gene annotation´.
6. In general, the authors report DEGs, but lists are missing with the actual gene identities. Please add these tables (can be suppl. data) .
For instance: ´Finally, 3817 DEGs were identified, 186 including 1341 up-regulated and 2476 down-regulated genes, in the aborted pistils compared 187 with the normal pistils (Fig. 1B, C).´
´….84 DEGs were enriched in the plant hormone signal transduction pathway….´
´…..we found that 68 DEGs were enriched in phenylpropanoid biosynthesis….´
Where are the tables?

Reviewer 2 ·

Basic reporting

See below.

Experimental design

See below.

Validity of the findings

See below.

Additional comments

The manuscript by Li and colleagues presents comprehensive transcriptomic and candidate gene analysis associated with pistil development of pumpkin plants. The authors used RNA sequencing (RNA-Seq) technologies to identify the potential key genes involved in the development of pistils. The experimental design is a typical comparison, and the aborted and normal pistils of pumpkin were used in this analysis. Several DEGs were validated using qRT-PCR. The authors also investigated the expression levels of ethylene candidate genes in different flower tissues and highlighted ERDBF3 and ERTF10 genes. Thus, one of the main findings is related to ethylene signal transduction genes as important players in the pistil developmental processes. The work may be technically sound, but I feel that the treatment of data is little bit shallow and do not go beyond descriptive statistical analyses. The study is too focused on the technology/RNA-Seq approaches and not enough on biology. I also found many problems in this manuscript and pointed them as follows.

Major concerns:
(1) Poor figures and legends
As a whole, the authors should improve the quality of the illustrations (Figures). In addition, the legends were too poor. For example, Figure 1: Enlarging the font size in x-y axes and labels is needed. It is difficult to check the scientific quality. The authors should describe the significance levels in Figures 1B and1 C.
Figure 2: The current legend was confused with that of Figure 3. You should explain software to generate this figure. What is RichFactor?
Figure 3: The font size is too small. The authors should use magenta/cyan instead of red/green. It is important from the viewpoint of "Colorblind Barrier Free".
Figure 4: The figure legends need more information about the bar plots. What are significant differential expressions? What do the bars define? Average, median, outliers, distribution, error bars? Did you perform any statistical tests? The authors should explain all the gene abbreviation.
Figure 5: See my comments to Figure 4.

(2) Identification of DEGs and functional enrichment analysis
I was not able to find any statistical tests (e.g., DESeq) to identify 3817 DEGs. Also you mentioned in L152-143: ”The |log2 (fold change)| > 1 and FDR<0.05 were used as significance cut-offs for the expression differences.” This is too arbitrary without any statistical tests. The correction method of p-values is not clear.
Please describe the parameters for transcriptome mapping, functional annotation, and enrichment analysis, such as TopHat2 and GOseq R. In the GO- and KEGG pathway enrichment analysis, did you use FDR correction for multiple testing problems?

(3) Biological significance and discussion
Table 2: Changes in most of candidate PAL genes are small but significant. Is this study, the authors mentioned about the role of PAL and ethylene candidate genes. It is absolutely wrong to put such argue that these genes are relevant for the biosynthesis and signal transduction pathways without any experimental evidence. It is highly important to conduct a silencing/mutant study of either any one of the mentioned genes to support your conclusion. Without experimental evidence and support, the authors cannot claim their role in the biological processes.

(4) Raw and processed data
The raw transcriptome data were deposited in the NCBI SRA under BioProject number PRJNA554766. This is great. However, the authors should add all the processed data such as the results including functional annotations and FPKM values of all transcripts as supplementary data (e.g. MS Excel file).

Reviewer 3 ·

Basic reporting

The ms by Qingfei Li et al. describes the results of a transcriptomic analysis on pumpkin pistils and the expression of some selected genes in different parts of the flowers in such species. The authors claim the importance the ethylene response and phenilpropanoid biosynthesis during pistil development. In general, the manuscript is well written, although it could be improved changing the structure of some sentences.
The introduction of the work indicates properly what is known about sex determination of flowers in cucurbits, which is a very early stage of flower development. As the topic of the work is pistil development the introduction should be illustrative about what is known for this in cucurbits and other species and how hormones impact on it.


-The main experiment of the work, the RNA seq, is not accessible and raw data have not been disposed in any database.

Experimental design

The main experiment of the work is an RNAseq. The authors have tested only two biological replicates in the experiment, aborted vs. “normal pistil”. They should describe in more detail what are aborted and normal pistils, and of course, indicate the developmental stage of the flowers at moment of the collection of the samples. I mean, for example flowers at anthesis? Before?... What I observe in the Figure1 A is a mix of pistils in different developmental stages. It could be also beneficial for unfamiliar readers to include the normal pistil for comparisons, and pictures of the flowers at the moment of collection. In addition, the origin of the abortion must be indicated in the text. Is due to the absence of ovules? defects in the cell division? Could this cause be responsible of part of the DEG observed? All this points should be addressed.

Validity of the findings

An important concern that I have is about how well annotated is the Cucurbita moschata genome, and how this information could be used in ontology analysis. In the list presented by the authors in supplementary figure 2, I see that they found a lot of ERF, but it make reference to a transcription factor family, and not to a function!
This point make the conclusions derived very weak.
For example AP2 and ANT genes (included in the list) belongs to this family but no relation with ethylene has been reported.
In addition, the authors named the different genes that they tested by Q-PCR with Arabidopsis thaliana names, as for example NPR1 or ERDBF3, without indicating the Cm locus. Are they sure that the genes at least are orthologous?
As a suggestion, I will reanalyse all this data carefully, and separating the up-regulated genes from the repressed genes.

---

## Round 0.2 · Minor Revisions

The reviewer agree on a substantial improvement of your paper. However, some remaining concerns still need to be addressed prior to publication.

·

Basic reporting

No comment.

Experimental design

No comment.

Validity of the findings

No comment.

Additional comments

The authors addressed all my comments.

Reviewer 2 ·

Basic reporting

See below.

Experimental design

See below.

Validity of the findings

See below.

Additional comments

I appreciate the authors' efforts to address the comments expressed during the previous round of reviews. The revised manuscript has much improved. However, I still had a minor comment about data reproducibility and transparency. The authors did not answer the following question in the previous review.
> I was not able to find any statistical tests (e.g., DESeq) to identify 3817 DEGs.

Reviewer 3 ·

Basic reporting

see below

Experimental design

see below

Validity of the findings

see below

Additional comments

I'll put my comments following the answers of the authors:

Experimental design
The main experiment of the work is an RNAseq. The authors have tested only two biological replicates in the experiment, aborted vs. “normal pistil”. They should describe in more detail what are aborted and normal pistils, and of course, indicate the developmental stage of the flowers at moment of the collection of the samples. I mean, for example flowers at anthesis? Before?... What I observe in the Figure1 A is a mix of pistils in different developmental stages. It could be also beneficial for unfamiliar readers to include the normal pistil for comparisons, and pictures of the flowers at the moment of collection. In addition, the origin of the abortion must be indicated in the text. Is due to the absence of ovules? defects in the cell division? Could this cause be responsible of part of the DEG observed? All this points should be addressed.

Agreed. According to the reviewer’s suggestion, we added more detail description of the aborted and normal pistils and indicated the developmental stage of the flowers at moment of the collection of the samples in the new manuscript.

---- I still see a problem using only two biological replicates in RNAseq experiments. The authors have added additional information about the nature of the sampling for the RNAseq, explaining that samples are collected from 5mm buds, and they refer to the fig1A. As I commented in my first report, samples in fig1A seems a mix of different developmental stages of pistils. Which one represents the pistil of a 5mm bud? 5mm bud are buds where pollination has not yet occurred? Is there any "normal" pistil to compare? Are there macroscopic differences between the two kind of pistils analyzed? The authors also indicated that one difference between aborted and normal pistil is the absence of ovules. Again, I wonder if part of the DEG observed could be related to this fact instead of proper pistil development. These points should be addressed and, at least, considered in the discussion.-------

Validity of the findings
An important concern that I have is about how well annotated is the Cucurbita moschata genome, and how this information could be used in ontology analysis. In the list presented by the authors in supplementary figure 2, I see that they found a lot of ERF, but it make reference to a transcription factor family, and not to a function! This point make the conclusions derived very weak. For example AP2 and ANT genes (included in the list) belongs to this family but no relation with ethylene has been reported.
In addition, the authors named the different genes that they tested by Q-PCR with Arabidopsis thaliana names, as for example NPR1 or ERDBF3, without indicating the Cm locus. Are they sure that the genes at least are orthologous?
As a suggestion, I will reanalyse all this data carefully, and separating the up-regulated genes from the repressed genes.

The Cucurbita moschata genome is annotated well. The ERF, AP2 and ANT in the new supplementary Table 4 were annotated as ethylene-responsive transcription factor in the NCBI blast and Cucurbita moschata genome. So, they have relation with ethylene.
According to the reviewer’s suggestion, the Cm locus of genes that were tested by Q-PCR have been added in the new manuscript. They are orthologous. Thanks for your suggestion.

Again I'm not agree with the consideration that all genes in the list are "ethylene signal transduction genes"or "ethylene-responsive factors". At least in Arabidopsis, AP2/ERF transcriptions factors are a huge family, that are involved in multiple developmental processes and respond to many abiotic a biotic stresses as well as multiple hormones, but this doesn't mean that all them respond to ethylene. If authors consider that genes included in the list are clearly ethylene response genes, please include a reference where this is stated.
Other point that I suggested but not considered by the authors was to perform the GO analysis separating Up-regulated and down-regulated genes. Many times these analysis make results clearer.

---

## Round 0.3 · Minor Revisions

Thank you for revising your manuscript.

The section editor made some additional comments you should address:

'1) The differential expression analysis was done incorrectly and needs to be corrected. Specifically the authors used FPKM as input to edgeR, but edgeR requires raw read counts. From the edgeR manual: "Note that normalization in edgeR is model-based, and the original read counts are not themselves transformed. This means that users should not transform the read counts in any way before inputing them to edgeR. For example, users should not enter RPKM or FPKM values to edgeR in place of read counts. Such quantities will prevent edgeR from correctly estimating the mean-variance relationship in the data, which is crucial to the statistical strategies underlying edgeR. Similarly, users should not add artificial values to the counts before inputing them to edgeR.

2) In addition, the study uses a replicate number for RNAseq differential expression analysis of 2, which is below standard in the field.'

In some parts, you could be more confident about your findings. In particular, I would suggest you change the following phrases that are undermining the contribution of your study:

Abstract:
"These results should provide a theoretical basis for further understanding of the mechanism"
- Why "should"? "Our study provides ..."

Conclusions:
"However, the key genes that lead to pistil abortion need to be further functionally validated, to clarify their mechanism in female flowers development and in sex expression of pumpkin." (BTW: should be "flower".
- This final phrase indicates that your study is incomplete since you did not verify/ validate your results. Please consider either deleting this sentence or writing it in a more positive sense; e.g. "Further studies will investigate ..."

---

## Round 0.4 · accepted · Accept

I appreciate the additional corrections of your manuscript. Congratulations!